# Establishment and Comparison of Pathogenicity and Related Neurotropism in Two Age Groups of Immune Competent Mice, C57BL/6J Using an Indian Isolate of Chikungunya Virus (CHIKV)

**DOI:** 10.3390/v11060578

**Published:** 2019-06-25

**Authors:** Jaspreet Jain, Vimal Narayanan, Ankit Kumar, Jatin Shrinet, Priyanshu Srivastava, Shivam Chaturvedi, Sujatha Sunil

**Affiliations:** 1Vector Borne Diseases Group, International Centre for Genetic Engineering and Biotechnology, New Delhi, Delhi 110067, India; jaspreet.jain@gmail.com (J.J.); drvimaln@gmail.com (V.N.); ankitkumar.bcas@gmail.com (A.K.); jatbioinfo@gmail.com (J.S.); priyanshu@icgeb.res.in (P.S.); 2Animal House Facility, International Centre for Genetic Engineering and Biotechnology, New Delhi, Delhi 110067, India; shivam@icgeb.res.in

**Keywords:** chikungunya virus, animal model, pathogenicity, C57BL/6J mice, disease progression

## Abstract

Chikungunya (CHIK) is a febrile arboviral illness caused by chikungunya virus (CHIKV) and has been identified in more than 60 countries across the globe. A major public health concern, the infection occurs as an acute febrile phase and a chronic arthralgic phase. The disease manifests differently in different age groups that can range from asymptomatic infection in the younger age group to a prolonged chronic phase in the elderly population. The present study was undertaken to evaluate strain-specific pathogenesis of ECSA genotype of CHIKV strains derived from clinical isolates in adult C57BL/6J mice model. The strain that was pathogenic and developed distinct acute and post–acute phase of CHIK infection was further evaluated for dose-dependent pathogenesis. Upon arriving on the optimal dose to induce clinical symptoms in the mice, the disease progression was evaluated across the acute and the post–acute phase of infection for a period of 15 days post–infection in two age groups of mice, namely eight weeks old and 20 weeks old mice groups. Biochemical, hematological, and virology attributes were measured and correlated to morbidity and linked neurotropism and limb thickness in the two age groups. Our results show that CHIKV exhibit strain-specific pathogenesis in C57BL/6J mice. Distinct dissimilarities were observed between the two age groups in terms of pathogenesis, viral clearance and host response to CHIKV infection.

## 1. Introduction

Chikungunya disease (CHIK) is caused by chikungunya virus (CHIKV), a positive single stranded RNA virus of genus alphavirus and family Togaviridae [1,2]. As commonly observed in other viral diseases, CHIKV infection may be asymptomatic or produce a variable spectrum of clinical manifestations, ranging from milder forms to severe and debilitating conditions [3]. Symptomatic chikungunya infection has been classified into three phases: Acute, post–acute, and chronic [4]. Pyretic or the acute phase of the disease lasts up to 2–7 days; symptoms generally include very high fever, nausea, diarrhea, headache, and rashes accompanied with severe arthralgia. Once the fever subsides, some of the symptoms such as arthralgia, headache, and restriction of joint movements may persist up to 21 days post–infection (dpi) and is generally referred to as the post–acute phase. A percentage of patients then graduate to the chronic or arthritic phase of the disease that is marked by severe joint swelling, joint restriction and polyarthralgia of varying degrees. Neuro-invasion by CHIKV causing viral encephalitis has also been observed, a rare complication recently reported observed mainly in the older patients [5,6,7]. In the post–pyretic or the post–acute phase, arthralgia continues leading to severe pain lasting for months till several years as reported by previous studies [8,9].

The extent of symptomatic and asymptomatic infections varies depending on patient age groups, circulating strain and possibly according to the prevalence of specific antibodies (Abs) developed against CHIKV infections [10]. Additionally, several studies have emphasized on the correlation of patients’ age to the severity of clinical manifestation of the disease [11,12,13,14,15]. Pathogenesis also is dependent on a variety of factors including type of vertebrate host, age of the host and virus strains [16,17], which makes studying the underlying mechanisms of disease pathogenesis more complicated. Currently, animal models used to study CHIKV pathogenesis are unable to adequately recapitulate the immunological response observed in patients [18] due the reasons as mentioned above. Haese et al. [19] outlined CHIKV disease mouse models into three major categories; the lethal neonatal models that develop encephalitis and offer an important system to study viral and host factors contributing to severe disease in neonates [20,21]. These models are used to test the efficacy of anti-CHIKV therapeutics owing to their high sensitivity towards CHIKV infection and replication [19]. Nonetheless, the immature state of their immune systems significantly limits its utilization to study chikungunya pathogenesis. The next category of animal models are the immune-compromised models, lacking in functional type I IFN receptor and associated components of the pathway are highly susceptible to CHIKV infection [20,22,23,24]. These mice models are used to study the involvement of type I IFN system in CHIKV pathogenesis. Due to the lack of a fully competent innate immune system, these mice display more severe muscle and joint pathologies. However, the accurate reflection of CHIKV pathophysiology in these models is a concern. Yet, they are used to study the efficacy of anti-CHIKV Abs and to evaluate the safety and efficacy of chikungunya vaccines [25,26]. The third category of mice models are the CHIKV arthritis/myositis models, those that are immune competent mouse models that are generally utilized to evaluate CHIKV specific vaccines and drug candidates [27,28,29] and for testing potential CHIKV inhibitors [30]. Also, these mice are used to investigate the persistence of CHIKV infection and its association with chronic disease and are the most appropriate models to study CHIKV pathogenesis [31,32,33]. 

Several studies have emphasized on the severity of infection and extended chronic phase that is age dependent [13,14,34]. In Indian outbreaks, the age of clinical cases of CHIK have been reported to be in the range of 11–55 years with a median age group of 35–45 years [15,35,36]. Furthermore, these studies have reported differences between the young adult and mature adult groups in their disease progression and disease resolution [15,34]. Earlier studies evaluating CHIKV pathogenesis have utilized 8 week old mice that is usually considered as young adults and its biological adulthood age corresponded to 15–21 years. However, the more relevant age group in mice studies to evaluate CHIKV pathogenesis in humans aged 45–50 years corresponded to around 20 weeks old mice [37,38]. This age group is considered as mature adults and have been used in studies evaluating arthritis along with 8 week old mice [39]. Further, a recent study performed during CHIKV outbreak in India also reported neurological complications associated with the CHIKV infection [40].

The present study was undertaken to address some of these aspects of CHIKV pathogenesis. We studied CHIKV disease pathogenesis in 20 weeks old mice that correlates to 40–45 human years [37]; compared it with the well-established 8 week old C57BL/6J mice model, and studied aspects such as virus strain-specific pathogenicity, effect of host age in CHIKV pathogenesis and the associated host response and related neurotropism. Dose-dependent viral pathogenicity of CHIKV strains obtained from clinical CHIKV isolates was tested in C57BL/6J mice. Upon arriving upon the pathogenic virus strain that developed distinct acute and post–acute CHIK infection phases and optimal dose of virus that induces symptoms simulating human disease, we proceeded to evaluate age dependent host responses in the mice at two age groups, i.e., 8 weeks and 20 weeks. The model developed in this study recapitulate CHIK disease severity in young and old age mice and aid in identifying risk factors associated with age and increased disease severity. This may potentially serve as a platform for testing vaccines in a more vulnerable age group. Furthermore, the model described in this study may be useful in determining age–specific differences in anti-CHIKV immune responses.

## 2. Materials and Methods

### 2.1. Animal Ethics Statement

C57BL/6J mice were originally purchased from Jackson Labs (Sacramento, CA, USA) and inbred locally at ICGEB. Work with infected animals was carried out in animal biosafety level (ABSL)-2 facility. All animal protocols were approved by ICGEB-Institutional Animal Ethics Committee (IAEC) [Approval Number: ICGEB/IAEC/08/2016/VBD-1(Extension), Approval Date: 15 September 2016]. All the animals were cared for and euthanasia was induced by placing each mouse in a CO_2_ regulated inhalation chamber with a calibrated gas controller as recommended by the ICGEB-IAEC. 

### 2.2. CHIKV Amplification and Quantification and Characterization

Lab amplified virus stocks were prepared as previously mentioned [15,41]. The virus was quantified using modified plaque assay for 96 well plate format was performed using Vero cells (ATCC^®^ CCL-81™). Briefly, starting at 1:100 the virus was double diluted till 1:102400. The virus was incubated on the cells for 1–2 h at 37 °C for virus adsorption. Thereafter, the viruses were removed, and wells were overlaid with 150 μL of 1% carboxymethylcellulose (CMC) prepared in sera free DMEM media (i.e., overlay media). The plates were incubated at 37 °C for 48–72 h at 37 °C with 5% CO_2_ and 75% humidity. Post incubation the cells were fixed with 10% formaldehyde before washing twice with 1× PBS and staining with 0.25% crystal violet (prepared in 30% methanol). The stained wells were washed with 1× PBS. Virus titers was calculated using the following formula:Plaque forming units (pfu) = (No. of plaques)/(Dilution × volume of virus)(1)

Further, viral characterization was performed using Viral nucleic acid (vRNA), isolated from viral stock using High Pure Viral Nucleic Acid Kit (Roche, Grenzach-Wyhlen, Germany) and estimation of viral copy number was done using the QuantiTect reverse transcription kit (Qiagen, Hilden, Germany) as per previously established protocols [15].

### 2.3. Determination of Optimum Dose to Establish CHIKV Infection in C57BL/6J Mice

Viruses of different concentrations were injected in 8 weeks and 20 weeks old C57BL/6J mice to determine the 50% mouse infection dose (MID_50_) by footpad injections [42]. Twenty four age matched mice were divided into six groups of eight mice each (4 male and 4 female mice) and infected with varying CHIKV PFU (Group 1: PBS mock, Group 2: Attenuated virus control (UV treated 1 × 10^8^ PFU), Group 3: 1 × 10^2^ PFU, Group 4: 1 × 10^4^ PFU, Group 5: 1 × 10^6^ PFU and Group 6: 1 × 10^8^ PFU). All mice infections were performed subcutaneously in the footpad of the left hind limb at a final volume of 50 μL. The mice were observed for changes in morbidity conditions by physical observations as per the Morton and Griffith Scale [43], that takes into consideration the changes in weight, physical appearance, clinical signs such as changes in temperature, cardiovascular, nervous and digestive responses and behavior changes to both unprovoked and external stimuli. Further, weight and temperature were measured using laboratory grade weighing scale and IR based thermometer respectively. Further, swelling in the limbs was measured using Vernier calipers to the limit of 0.1 cm and standard deviation of 0.002 cm. The study was done for 15 days to study the acute and post–acute phase of CHIK.

### 2.4. Evaluation of Disease Progression in C57BL/6J Mice Using CHIK/DEL/2010/01 CHIKV Isolate

C57BL/6J mice were evaluated to ascertain CHIKV infection in 8 and 20 weeks old C57BL/6J mice. 1 × 10^6^ PFU CHIKV was injected in the footpad of the hind limbs of the animals, a total of 4 mice per day of the study for 15 dpi. Mice were observed for changes in the clinical parameters such as morbidity, weight, temperature, and swelling of the limbs. Further, mice were sacrificed, and organs (joint, brain, muscle, liver, skin, intestine, bone marrow, blood/serum, heart, thymus, lungs and kidney) were recovered and used to evaluate day-wise viral load in the animals. Splenocytes were isolated from mice spleen, pooled and used for further experiments. Euthanasia criteria was decided based on the morbidity score recorded in accordance with the Morton and Griffith scale [43]. Concurrent experiments were performed for 8 weeks and 20 weeks old mice.

#### 2.4.1. Day-Wise Viral Load Estimation

Organs were harvested from the euthanized infected animals. Every organ was weighed homogenized with PBS in the weight to volume ratio of 1:5 and lysed using tissue lyser (Qiagen, Hilden, Germany). 200 μL of this homogenate was used for viral nucleic acid (vRNA) isolation using High Pure Viral Nucleic Acid Kit (Roche, Grenzach-Wyhlen, Germany) and estimation of viral copy number was done using the QuantiTect reverse transcription kit (Qiagen, Hilden, Germany) as per previously established protocols [15].

#### 2.4.2. Blood Sampling Method and Sample Handling

Retro-orbital blood samples were collected from the right retroorbital plexus of anesthetized mice. Blood from the whole body was collected by heart puncture. Blood samples were divided into two equal volumes and deposited in serum separator tubes as well as blood collected tube with EDTA (Microtainer, Becton–Dickinson, Franklin Park, NJ, USA). Serum hemolysis was evaluated by direct observation. 

#### 2.4.3. Clinical Chemistry Parameters

Whole blood samples were used to determine the complete blood profile of the infected mice. Additionally, serum activity for the presence of aspartate transaminase (AST), alanine transaminase/alanine aminotransferase (ALT), alkaline phosphatase (AP) and rheumatoid factor (RF) was evaluated using an automated analyzer according to the manufacturers’ instructions. Standard controls were run before each determination, and the values obtained for the different biochemical parameters were reported. 

#### 2.4.4. Estimation of the Plaque Forming Units

Plaque forming units in the sera of the infected mice were estimated in a day-wise manner using previously established protocols as aforementioned.

#### 2.4.5. Estimation of Presence of Binding Abs

To detect titers of CHIKV specific IgM and IgG Abs, indirect ELISA using purified CHIKV as coating antigen was performed using previously established protocols [15]. Briefly, purified CHIKV particles (2.5 μg/mL/5000 virus particles, 100 μL/well) were coated in coating buffer (1× PBS with 2% FBS) in microtiter plates and left undisturbed overnight at 4 °C. Plates were washed, blocked and two-fold serially diluted mouse samples starting at 1:100 were incubated on the plates at room temperature for two hours with gentle rocking. After washing, the plate was developed using anti-mouse IgM HRP and anti-mouse IgG HRP respectively followed by TMB as substrate. Previously well characterized samples with confirmed presence of CHIKV specific IgM and IgG Abs respectively were pooled and used as a positive control to obtain a linear curve for absorbance and dilution. Initial titers and mid-point titers for IgM and IgG Abs respectively were calculated.

#### 2.4.6. Neutralization Status of the Binding Abs

The neutralization capacity of infected sera samples from mice was analyzed by plaque reduction neutralization test (PRNT). The neutralization assay was performed on Vero Cells using modified protocols [44]. Briefly, sera samples were heat inactivated at 56 °C for 30 min and then two-fold serially diluted sera (starting at 1:50) was incubated with 50 PFU CHIKV for 1hr at 37 °C. After incubation, Vero cells were infected with the virus and antibody mixture and incubated for an additional 36 h. Post 36 h, cells were fixed and the neutralizing (NT) Abs were measured as the percentage neutralization in viral plaques formation compared to the virus alone wells. The results were calculated as PRNT-50. All samples with PRNT-50 at 1:50 or below were scored as non-neutralizing. 

#### 2.4.7. Cytokine and Chemokine Estimation

Levels of CCR1, CCR2, IL-4, BST and IFN-gamma were estimated in the splenocytes of the infected mice with FACS Canto machine; BD Biosciences using differently conjugated Abs as per the standardized BD protocols. Cell Quest software (Becton-Dickinson, Franklin Lakes, NJ, USA) was used to analyze the presence/absence of these cytokines at 3, 6, 9, 12 and 15 dpi. All experiments were done in triplicates and coherent data were used for inferences. Splenocytes of uninfected male and female mice was used as a control and the references were decided according to these sample. On the other hand, levels of IL-1β, IL-6, TNF-α, GM-CSF, CCL2 (MCP-1), VEGF, ICAM-1, CCL5 (Rantes) were analyzed with Luminex-200 system (Bio-Rad Laboratories, Hercules, CA, USA) using Magnetic Luminex screening assay by RND systems. Sera samples were diluted 1/2 and assessed in duplicate according to the manufacturer’s instructions. Assays were analyzed using on the Bio-Plex Manager software, Standard Edition (Bio-Rad Laboratories, Hercules, CA, USA).

#### 2.4.8. Immuno-Histopathology and Confocal Microscopy

Immuno-histopathology was done for infected mice brain and joint tissue. Briefly, paraffin-embedded, 5-μm sections of brain and joint tissues were stained using an automated processor using monoclonal anti-CHIKV antibody (3585) (Abcam, Cambridge, UK) at a dilution of 1:200, followed by biotin–labeled goat anti-mouse secondary antibody at a dilution of 1:800 and horseradish peroxidase-labeled streptavidin and developed using 3-Amino-9-Ethylcarbazole substrate system (Abcam, Cambridge, UK) to give red color during confocal microscopy Immunohistochemically stained sections were evaluated under confocal microscopy by scoring the slide for density of the stained cells in lesioned areas and the morphology of the infected areas. Presence of virus antigens in joint and brain tissues was further evaluated. 

### 2.5. Statistical Analysis

To identify the optimal number of mice, a power analysis for determining sample size (SPSS 10.0, SPSS, Chicago, IL, USA) was conducted with an alpha value of 0.05 and a power of 90%. Previously published data for C57BL/6J mice was used to establish an expected difference in means and an expected SD for each biochemical parameter to perform this calculation. For all the other parameters and time point, the group mean, SD was calculated. Statistical analyses were performed using GraphPad Prism 6 software. The Grubb test (GraphPad Software, La Jolla, CA, USA) was run for outlier detection. Outliers were removed, and group means, SD was recalculated. To compare data between groups (GraphPad Prism 6), the Wilcoxon signed-rank test was used when conditions of normality and equal variance were met. The Student t test was used with Welch correction when unequal variances were detected. When the normality test failed, a 2-tailed and exact Mann-Whitney rank sum test was used. Differences were considered statistically significant at a *p* value of less than 0.05. Correlation was determined using Spearman’s rank-order correlation analysis at confidence interval of 95% and R^2^ values at *p*-value < 0.005 were considered significant.

## 3. Results and Discussion

Studies evaluating the mechanisms of CHIKV pathogenesis are limited mainly owing to the lack of a suitable animal model that studies the various aspects of pathogenesis as observed in humans [18,20,45]. In other arboviruses, attempts have been made to develop a strain-specific pathogenesis model in an otherwise refractory animals in order to study certain mechanistic aspects of the virus and the disease [46]. In humans, CHIK presents with a varying level of clinical symptoms and a variety of features such as viral load, neutralizing capacity of the virus, patient age, joint movement restrictions and joint swelling, that may also plays important roles in disease progression [47,48]. Recent studies from our lab have attempted to address some of these concerns in humans during CHIK outbreaks in the country [15,35]. In the present study, we evaluated 15 CHIKV strains, collected from patients during the CHIKV outbreaks, for their pathogenicity in terms of CHIKV induced morbidity and associated arthralgia, common symptoms associated with CHIK. In this study, we show that CHIKV isolates exhibit strain-specific pathogenicity that was dose-dependent, an important aspect to be considered for future animal pathogenicity experiments. We further used the CHIKV pathogenic strain that developed distinct acute and post–acute disease phases to evaluate the differences in host response owing to age of the host during both acute and post–acute phase of the disease. We observed older mice responded differently to CHIKV infection as opposed to younger mice in terms of virus dose-dependent disease severity, virus replication kinetics, cell-mediated immune responses, development of binding Abs and their neutralization capacities etc. We further resolved to check for any plausible reason for neurological complication associated with CHIKV infection, a rather rare but lethal disease outcome observed in the old age patients, mainly in the previous Indian outbreaks [40]. The details of these are being discussed in detail in the sections below.

### 3.1. CHIKV Strains Differ in Their Pathogenic Potential in C57BL/6J Mice

CHIKV strains at a final concentration of 1 × 10^8^ pfu/50 μL were injected subcutaneously in the hind limbs of 8 weeks and 20 weeks old C57BL/6J mice. CHIKV strains showed differences in their pathogenesis in both the age groups and were assayed in terms of morbidity, limb thickness and body weight and survivability (Appendix A). Evaluation of the 15 CHIKV strains revealed that amongst the most pathogenic strains, strain CHIKV#01 was most optimal with distinct acute and post–acute phase based on most common disease markers; morbidity and limb thickness. On the basis of our observations, strain CHIKV#01 was further used for dose optimization. 

### 3.2. Pathogenicity of CHIKV#01 Strain Is Dose-Dependent

Since the scope of this study was to evaluate acute to post–acute phase pathogenesis of CHIKV without medical interventions, our study was designed to be terminated in 15 days as beyond this point medical intervention was suggested by IAEC. We evaluated the pathogenicity of CHIK#01 in a day-wise manner for duration of 15 days using multiple concentrations of the virus, concurrently in 8 weeks and 20 weeks old mice. Dose-dependent analysis of survival of mice owing to virus concentration of CHIKV#1 revealed that MID_50_ for both 8 weeks and 20 weeks mice was between 10^6^ and 10^8^ PFU (Appendix A). 

Further analysis of morbidity in the 8 weeks old animals infected with CHIK#01 revealed that a dose of 10^6^ and 10^8^ PFU was effective in inducing morbidity (morbidity score 5–10) in both male and female genders. The animals infected with 10^2^ and 10^4^ PFU were found normal without any signs of disease manifestations (morbidity < 5). In case of animals infected with 10^8^ PFU, morbidity score reached as high as 6.5 and 8 when infected with 10^6^ and 10^8^ viruses respectively at 15 dpi with an intermittent drop at 12 dpi (Figure 1). Whereas, in case of 20 weeks old mice animals reached a maximum score of 10 when infected with either 10^6^ and 10^8^ viruses at 15 dpi, post which the animals either died or were terminated as per the IEAC recommendations (Figure 1). 

We further evaluated limb thickness to score day-wise pathogenicity in a dose-dependent manner. Limb thickness at the site of infection was measured at 0, 3, 6, 9, 12 and 15 dpi in 8 weeks and 20 weeks mice. Animals infected with 10^2^ and 10^4^ PFU showed no change in the limb thickness when compared to the control group in both the age groups. In case of 8 weeks old mice, >1× change in limb thickness (*p* value < 0.05) were observed at 9 dpi in the mice infected either with 10^6^ or 10^8^ PFU. Whereas, in 20 weeks old mice infected with 10^6^ and 10^8^ virus particles, >1× change in limb thickness was observed at 9 dpi and 6 dpi respectively (Figure 1). 

Based on the above observations, virus PFU 10^6^ induced optimal pathogenicity and was used to study virus pathogenesis in 8 weeks and 20 weeks old C5BL/6J mice for duration of 15 dpi. 

### 3.3. Hematological Markers Demonstrate Severity of Acute Infection in Older Mice and Development of RA Factors in the Post–Acute Phase of Infection

As a first step to study CHIKV pathogenesis, we evaluated the various hematological and biochemical parameters in the infected mice of both age groups. Table 1 summarizes the complete blood parameters of infected 8 weeks and 20 weeks old mice, respectively, at various time points of the study. Our results show that for 8 weeks as well as for 20 weeks old infected mice, the median values of total leucocyte count (TLC), differential leucocyte count (DLC) of neutrophils, eosinophils and monocytes increased starting 6 dpi and 3 dpi for 8 weeks and 20 weeks old mice respectively, which may be due to the establishment of viral infection and inflammation. Interestingly, the TLC level went back to the normal in case of 8 weeks old mice while in 20 weeks old mice, TLC levels were increased until the termination of the study. The platelet count was higher than the normal range in both 8 weeks and 20 weeks old mice indicating thrombocytosis in the infected animals. Further, we observed differences in the hematological makers in the older infected mice. Reduced hematocrit and increased MCH, MCV and MCHC were observed throughout in the 20 weeks old mice. Whereas, none of the 8 weeks old animals exhibited any such change (Table 1a,b). Upon physical inspection, spleen enlargement was also observed in most of the older infected mice after 6dpi (83.33%, 30/36 animals) (figure not shown). 

Thereafter, sera samples of both 8 weeks and 20 weeks old mice were checked and the statistical data of biochemical parameters for sera from infected mice at various time points of the study are presented in Table 2. In 8 weeks, old mice, AST and ALT in the sera samples were higher than the normal range starting 6 dpi and it plateaued out at 12 dpi. While for 20 weeks old mice increased levels of AST and ALT were observed starting 3 dpi and it remained so until the termination of the study. Increased levels of AST and ALT indicate the liver associated symptoms during CHIKV infection as reported in other studies, both in patients and in non-human primates [49,50,51]. 

In the present study, we were also interested in understanding the overlap of CHIKV induced arthritis with that of rheumatoid arthritis (RA). In this respect, rheumatoid factor (RF) was checked, we observed that RF was absent in 8 weeks old mice at all the time points while in 20 weeks mice, 50% (*n* = 2) were positive for RF in sera at 12 dpi. These mice showed increase in the limb thickness (mainly in the hind limbs and in the fore limbs to some extent), and exhibited signs of self-mutilation (at the site of infection) (data not shown) indicative of extreme pain. 

Several studies in patients >35 years of age, studying correlation between RF and age of patients have also reported similar outcomes of incapacitating arthritis during post–acute phase of CHIKV infection [52,53] and an overlap between RA and seropositive spondylarthritis [54] as observed in the older mice in this study. However, in younger adult age group, levels of RF were not necessarily elevated in the acute and chronic phases of the diseases, which may indicate inflammatory reaction induced arthritis emerged post–CHIKV infection [8].

### 3.4. CHIKV#01 Persists Longer and Have Severe Disease Progression in Older Mice as Compared to Younger Mice

Sera samples were collected from the infected mice every third day and analyzed for the presence of viruses circulating in the blood. In case of 8 weeks old mice, we observed replicating virus particles only until 9 dpi (Figure 2), thus corroborating with previously published studies that explained CHIKV as a self-limiting infection [55]. Whereas in the 20 weeks old mice, pfu was observed until day 15 dpi, emphasizing on the differences in disease presentation in distinct age groups (Figure 2). Day-wise analysis revealed that at 3 dpi and 6 dpi, similar pfu was observed in both age groups (mean pfu at 3 dpi = 7.03 × 10^6^ and 8.28 × 10^6^ in 8 weeks and 20 weeks old infected mice respectively, mean pfu at 6 dpi = 4.43 × 10^6^ and 4.5 × 10^6^ in 8 weeks and 20 weeks old infected mice respectively). However, after 6 dpi, viral titers reduced in 8 weeks old mice but not in 20 weeks old mice, hinting towards the longer persistence of viruses in the 20 weeks old mice. Owing to the study design, we could not prolong the study beyond 15 days to check the time of virus persistence in the older age group. Nevertheless, the results suggest on age-specific differences in disease progression.

We then investigated the kinetics of CHIKV replication in body tissues of 8 weeks and 20 weeks old mice at 3, 6, 9, 12 and 15 dpi (or before, depending on the time of death). In case of 8 weeks old mice, at 3 dpi, viral load was highest in the spleen followed by skin, muscle liver, kidney, intestine and bone marrow. Post–6dpi, there were substantial differences in viral load amongst the various tissues. While the viral load increased in the joint and brain tissues till 12 dpi, and then decreased at 15 dpi, a gradual decrease was observed in the other tissues 6dpi onwards and reached below detectable levels by 9 dpi (Figure 3). The 20 weeks old infected mice revealed similar changes in the CHIKV replication kinetics and disease progression, but in exaggeration. We observed highest number of virus molecules in spleen, muscle and skin followed by kidney, liver, intestines, bone marrow, joint and brain at 3 dpi. Viral load increased in joint, brain and muscles at 6 dpi, while a decline in virus load was observed in all the other organs and reached below detectable levels by 12 dpi. As seen in the 8 weeks old mice, a continuous increase in the viral load was observed in the joint and brain tissues of the 20 week old mice. 

Documentation of the day-wise viral load in various organs of 20 weeks old CHIKV infected mice is present in Figure 3. The limit of detection in the experiment was 10^3^ vRNA molecules. 

Based on the above findings, we further sought to evaluate the day-wise changes in the status of limb thickness at the site of infection, one of the most common features of CHIKV infection (Figure 4a) and observed more than 2× limb thickness increase post–6 dpi for both 8 weeks and 20 weeks old mice. Maximum limb thickness in 8 weeks old mice was observed at 12 dpi, a slow but gradual reduction in the limb thickness was observed thereafter (Figure 4a). On the other hand, a continuous increase in the limb thickness of 20 weeks old mice was observed until the termination of the study (maximum increase in limb thickness = 3.6×) (Figure 4a). Further, correlation between limb thickness and the presence of viral load was observed to be positively correlated in both 8 weeks (Figure 4b) and 20 weeks old mice (Figure 4c) but was significant only in case of the latter (R^2^ = 0.6721, *p*-value > 0.5 for 8 weeks old infected mice and R^2^ = 0.4717, *p*-value = 0.016 for 20 weeks old infected mice).

Further, CHIKV replication kinetics in the brain tissues in 20 weeks old mice prompted us to look into neuro-invasion of the virus, a rather uncommon symptom of CHIKV infection. We hypothesized that presence of CHIKV in the brain tissue could be triggering the morbidity changes observed in the present study. Similar cases of neuro-invasion have been observed in case of other arboviral infections such as West Nile Virus (WNV), Japanese encephalitis virus (JEV) and Tick-borne encephalitis virus (TBEV) that demonstrated the association of virus load with CNS infections in causing significant morbidity and mortality in humans [56,57]. To test this hypothesis, we measured morbidity at days 0, 3, 6, 9, 12 and 15 dpi for both 8 weeks and 20 weeks old mice. We observed significant morbidity changes (morbidity score > 5) starting at 6 dpi for 8 weeks old mice, thereafter, a gradual increase in morbidity was observed with an intermittent dip at 12 dpi (*p*-value < 0.005 using wilcoxon signed-rank test and compared to 9 dpi) (Figure 5a). Whereas, in case of 20 weeks old infected mice the changes in morbidity started at 3 dpi and it increased until the termination of the study. Both 8 weeks and 20 weeks old mice showed reluctance to move and changes in behavior towards external stimulus was observed post–6 dpi. In case of 20 weeks old mice, some animals showed signs of self-mutilation caused due to severe swelling and pain as aforementioned, at this point a high morbidity score was observed and the animals were euthanized as per the IEAC recommendations. 

We also observed that in case of 20 weeks old infected mice increase in morbidity was found to be directly proportional to the increase in the viral copy number in the brain (Figure 5c) (R^2^ = 0.5695, *p*-value = 0.0001), whereas, no correlation between the two was recorded in case of 8 weeks old mice (Figure 5b). Day-wise analysis of the viral presence in brain demonstrated that CHIKV infection in the brain preceded symptoms of morbidity in the infected mice. Similar finding have been documented in the previous studies in other arbovirus infection; neurological complications caused due to virus infection was correlated to severe morbidity or death and the severity of virus infection was reported to be dependent on virus virulence, level of viremia and maturity of the infected neuron [58,59].

### 3.5. Disease Severity Is Correlated with Changes in Cell–Mediated Immunity in Older Mice

CHIKV infection imparts both antibody–mediated as well as cell–mediated immune responses that have a direct bearing on disease progression to the post–acute and the chronic phase of the disease [48]. We evaluated some molecules of the cell–mediated immune system that are known to participate during CHIKV infection (Table 3). In the acute phase of the disease, i.e., between 1 to 6 dpi, we observed increased levels of BST-2, TNF-α and IL-1β. While in the post–acute phase of the disease (7 dpi–15 dpi), we observed increased level of IL-4, GM-CSF, MCP-1, RANTES and RANKL and the presence of CCR2 to some extent. Interestingly, IFN-α, IL-1β, IL-6 and BST-2 level were high during both acute as well as post–acute phase of the disease. 

BST-2 was present more in 8 weeks old mice as in comparison to the 20 weeks old mice; this molecule has been known to induce anti-viral role in vivo as it restricts the budding of virus particles from the infected cells [60]. It should be noted that in case of severe acute infections along with the other cytokines, CCR1 and CCR2 are also present ascertaining virus infiltration into the brain tissues and is also upregulated in case of virus dependent arthritis [61]. One might speculate that early infection in several organs leads to recruitment of monocytes/macrophages and this macrophage infiltration is regulated under the control of MCP-1 (CCL-2)/CCR2, a feature of damaged tissues [61] with a probable activation by NK and/or T cell-derived IFN-γ [62]. In our study, we further observed that inflammatory effectors IL-4, IFNγ were expressed in the splenocytes while IL-6 and TNFα were expressed in the sera of the infected mice in both acute and post–acute CHIKV infection. Previous studies have also demonstrated that clinical manifestations may result from excessively activated macrophages releasing pro-inflammatory mediators such as IL-6 and, to a lesser extent, TNF-α [63]. Once the viruses have infiltrated the joint or muscle, the macrophages are activated and regulate the local Th1/Th2 balance as a function of their own activation status (classical/M1 or alternative/M2) [64]. Our study showed that VEGF, a molecule that participated in M1 effector activity was present only in the acute phase time points and GM-CSF that was part of the M2 effectors was present exclusively in post–acute phase time points, thereby making us hypothesize that these molecules may modulate M1/M2 balance during disease progression [65]. These evidences hint towards a probable M1 to M2 switching in the post–acute phase of the disease [65]. Our observations also suggest that the post–acute course of CHIKV disease is caused by continuing inflammatory responses of IL-1β, IL-6 and IL-4 associated with persistent virus, rather than by virally induced autoimmunity, as is also observed in case of patients [49,66]. 

### 3.6. Neutralizing Role of Binding Abs (Anti-CHIKV IgM and Anti-CHIKV IgG)

Prior studies have established that not all binding Abs made during an active infection are neutralizing and that neutralization ability can mature over time [15,67]. In the present study, we studied binding Abs (IgG and IgM) and their neutralization status in 8 weeks and 20 weeks old infected mice. The overall differences in the neutralizing ability of binding antibodies of 8 weeks and 20 weeks old mice is depicted in Appendix A. Further, initial point analysis for the development of CHIKV specific IgM Abs in 8 weeks and 20 weeks old mice started at 3 dpi although at varying concentrations (Figure 6a). IgM dependent CHIKV neutralization was observed in 8 weeks old infected mice (R^2^ = 0.8840, *p*-value < 0.0001) (Figure 6b). In case of 20 weeks old infected mice no such correlation was observed (R^2^ = 0.2253, *p*-value > 0.05) (Figure 6c).

We further studied the development CHIKV specific IgG Abs in both 8 weeks and 20 weeks old mice. Mid-point analysis revealed that in case of 8 weeks old mice development of anti-CHIKV IgG Abs started at 6 dpi (Figure 7a). On the other hand, in case of 20 weeks old mice, we observed the development of anti-CHIKV Abs at 9 dpi. The levels of IgG Abs were significantly lower than that observed in 8 weeks old mice but were maintained until the termination of the study (Figure 7a). A positive correlation between IgG titers and neutralization capacity of the sera samples was observed in case of both 8 weeks as well as 20 weeks old infected mice (R^2^ = 0.23, *p*-value < 0.05) and we observed that neutralization of CHIKV started at 6 dpi in 8 weeks old mice (Figure 7b). Whereas, in case of 20 weeks old mice, CHIKV neutralization was observed starting 9 dpi (Figure 7c), A slight positive correlation was observed between neutralization and IgG titers (R^2^ = 0.5460, *p*-value = 0.003 for 8 weeks old mice and R^2^ = 0.8471, *p*-value < 0.0001 for 20 weeks old mice) but the overall neutralization exhibited by 20 weeks old sera samples was lesser than that of the 8 weeks old infected mice for all the time points of the study. Based on the above results, we hypothesize that the neutralization status of 8 weeks old mice could be due to the complement of both IgM and IgG Abs during the initial days of infection, leading to early virus clearance (Appendix A). Whereas, in case of the older mice, the neutralization capacity of the sera samples was mainly due to the IgG Abs that were primarily produced during the post–acute phase of the disease and that too in lesser quantities resulting in delayed virus clearance (Appendix A).

## 4. Conclusions

This study was initiated to provide an in-depth analysis of CHIKV infection in an adult mouse model that could capitulate the pathogenesis pattern of CHIKV in the average age of human infection. Using a virus isolated from the recent Indian CHIKV outbreak (CHIK#01), we established infection in C57BL/6J mice and compared clinical and biological features of the infection in two different age groups of mice. Our results taken together show that CHIKV pathogenicity is strain, dose as well as age specific and these aspects need to be considered when studying virus replication and disease progression in any model organism. In our study, analyses of disease progression reveal that in the acute phase of infection, peak viremia, biochemical and hematological analyses demonstrate: (a) Increased levels of TLC indicating active infection in mice and elevated liver AST and ALT enzymes, sera AP; (b) high levels of CHIKV RNA in the brain, spleen, liver and skin, and to a lesser extent in joints, and muscle tissues and many of these tissues are also affected during human disease; (c) further correlation analysis revealed that morbidity and CHIKV replication in brain were associated indicating the involvement of CNS during the active stage CHIKV infection due to CHIKV infiltration in the mice brain during severe chikungunya infection; (d) the results demonstrated the importance of cell–mediated immune responses and we observed increased levels of IFN-α, BST-2, TNF-α, VEGF, IL-1β and IL-6 in the acute phase mice sera sample. Interestingly, the involvement of CCR1 and CCR2 were observed only during the severe stage of CHIKV infection. Additionally, antibody based CHIKV neutralization was also checked and samples did not show the presence of significant neutralization during the acute phase of the disease.

Further, the post–acute phase was characterized by (a) normalization of leukocyte counts by 10–15 dpi, and the presence of RF in the older mice post–severe–acute CHIKV infection; (b) pronounced arthralgia and macrophage infiltration in the joint; (c) increase vRNA levels in joint, and muscle. Further correlation analysis revealed that arthritis like symptoms caused may be because of inflammatory response of CHIKV caused due to the presence of virus particles in the joint tissues; however, in some cases during severe acute CHIKV infection, RF was also present indicating overlap between CHIKV induced arthralgia and rheumatoid arthritis; (d) cell–mediated immunity adversely affecting older aged mice; CMI analysis revealed increased levels of RANKL, RANTES, IL-6 and IL-1β all indicating the presence of CHIKV induced arthralgia; (e) Antibody-based neutralization assays demonstrated that CHIKV neutralization was due to the development of IgG Abs only in the younger mice, whereas older mice were not able to naturally neutralize CHIKV until the post–acute phase of the disease.

The present study had a few caveats. The study design did not permit us to extend the study beyond 15 days and so we could not evaluate viral presence beyond this period, especially in the older age group. Although the study explained the strain, dose and age specific differences of CHIKV isolates in the C57BL/6J mice, the study could have benefitted more if old age mice were also taken to evaluate geriatric age group during CHIK infection, considering CHIKV infects the geriatric age group (>65 years) in a more severe manner. Another aspect that would have been informative would have been studying if there were any mutational differences amongst CHIKV strains or even within the strain that could have led to mice adaptation. Such studies however require study design in an elaborate scale to provide meaningful information. In spite of these caveats, the study has provided important evidences regarding CHIK pathogenesis that could benefit the scientific community.

## Figures and Tables

**Figure 1 viruses-11-00578-f001:**
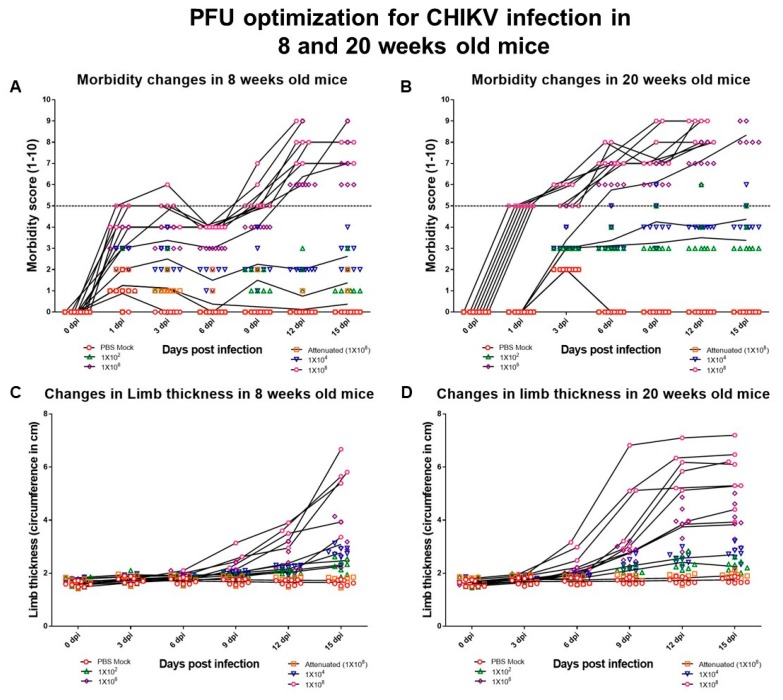
Dose-dependent pathogenesis of CHIKV#01. (**A**,**C**) Changes in morbidity and limb thickness post–virus infection at varying PFU in 8 weeks old mice. (**B**,**D**) Changes in the morbidity and limb thickness post–virus infection at varying PFU in 20 weeks old mice. Eight mice were taken each group. Error bars in all the figures above depict standard deviation. The level beyond which the changes were significant is represented in the figures.

**Figure 2 viruses-11-00578-f002:**
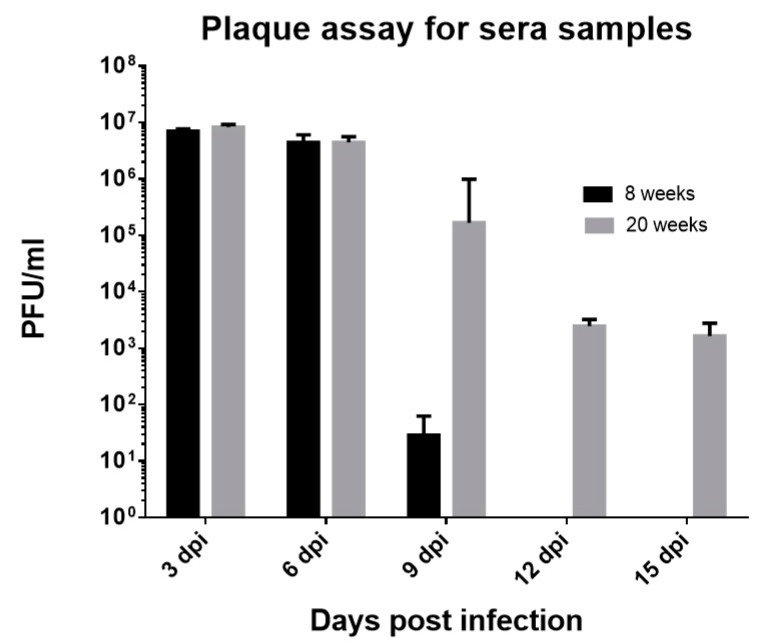
Day-wise presence of mean replicating virus particles in the 8 weeks and 20 weeks old mice post–CHIKV#01 infections. Four mice were taken in each group. Error bars in the figure depict standard deviation.

**Figure 3 viruses-11-00578-f003:**
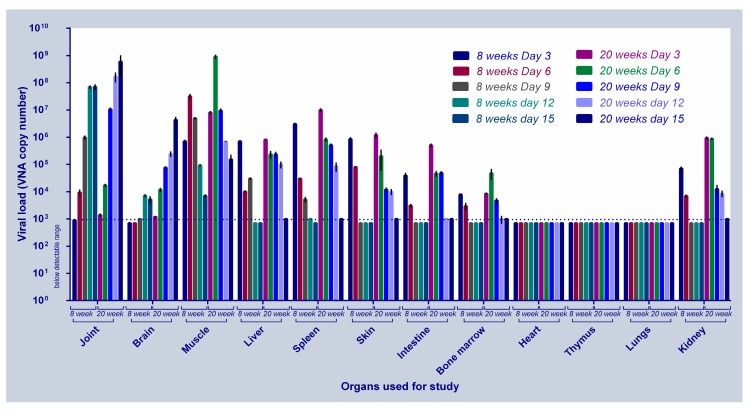
CHIKV replication kinetics in 8 weeks and 20 weeks old infected. Mean value of the vRNA copy number is represented for all the organs at various time points of the study. Level of detection is indicated as dotted line. Four mice were taken in each group. Error bars represent Standard error of the mean.

**Figure 4 viruses-11-00578-f004:**
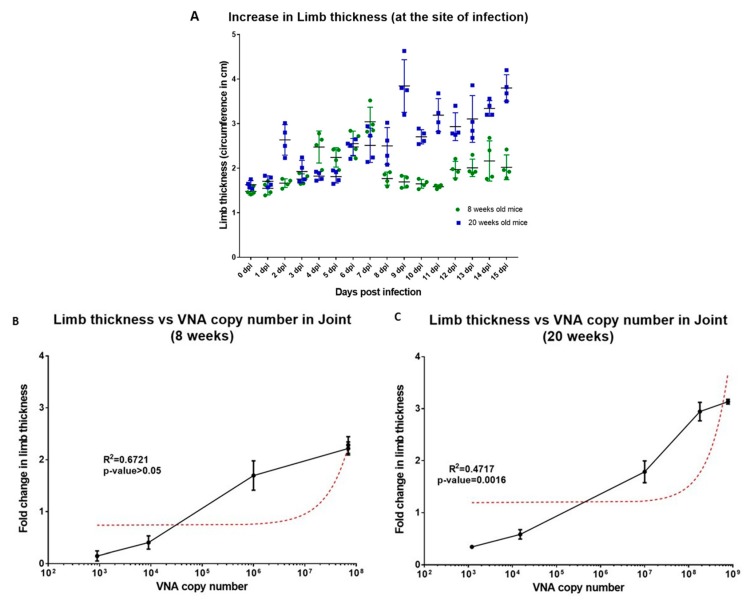
Limb thickness in mice correlates with Viral replication in the Joint tissues. (**A**) Fold change in the limb thickness in 8 weeks and 20 weeks old mice for a period of 15 days. Four mice were taken each group. (**B**,**C**) Correlation of fold change in limb thickness with the vRNA copy number present in the joint tissues of the CHIKV infected 8 weeks and 20 weeks old mice. Error bars in figure depict standard deviation. Spearman’s rank-order correlation analysis at confidence interval of 95% and R^2^ values at *p*-value < 0.005 were considered significant.

**Figure 5 viruses-11-00578-f005:**
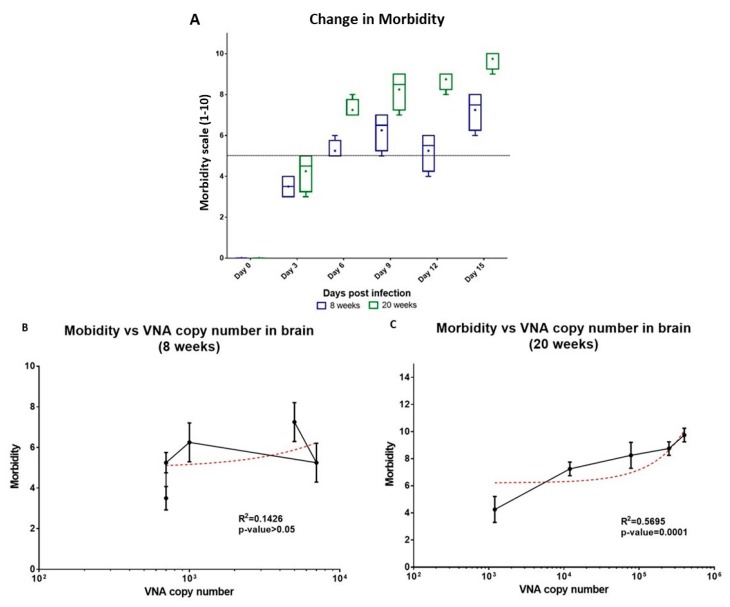
Morbidity in mice correlates with viral replication in brain. (**A**) Comparison of morbidity in 8 weeks and 20 weeks old mice for a period of 15 days. Four mice were taken in each group. (**B**,**C**) Correlation of morbidity with the vRNA copy number present in the brain of the CHIKV infected 8 weeks and 20 weeks old mice. Error bars in figure depict standard deviation. Spearman’s rank-order correlation analysis at confidence interval of 95% and R^2^ values at *p*-value < 0.005 were considered significant.

**Figure 6 viruses-11-00578-f006:**
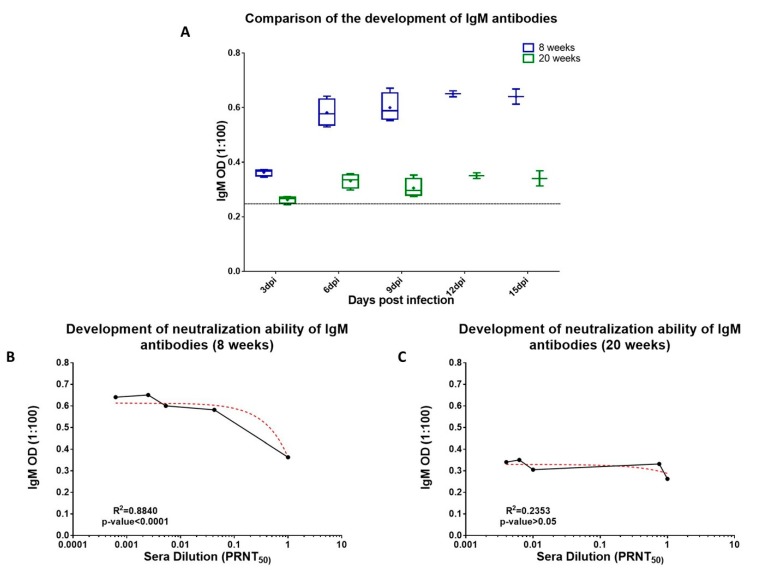
Status of the development of IgM Abs and their neutralization capacities in 8 weeks and 20 weeks old mice. (**A**) Comparison of day-wise development of IgM Abs between 8 weeks and 20 weeks old mice. Four mice were taken each group. (**B**,**C**) Correlation between development of IgM Abs and sera neutralization in 8 weeks and 20 weeks old mice. Spearman’s rank-order correlation analysis at confidence interval of 95% and R^2^ values at *p*-value < 0.005 were considered significant.

**Figure 7 viruses-11-00578-f007:**
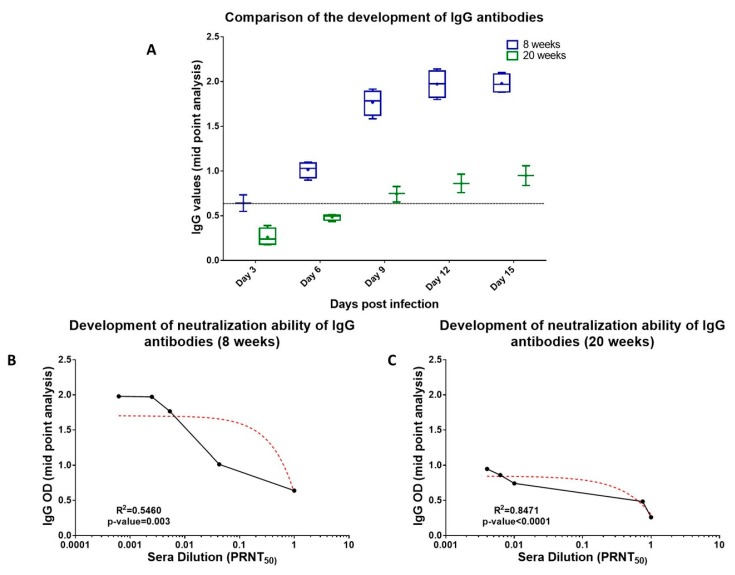
Status of the development of IgG Abs and their neutralization capacities in 8 weeks and 20 weeks old mice. (**A**) Comparison of day-wise development of IgG Abs revealed that IgG Abs in 8 weeks and 20 weeks old mice. Four mice were taken each group. (**B**,**C**) Correlation between development of IgG Abs and sera neutralization in 8 weeks and 20 weeks old mice. Spearman’s rank-order correlation analysis at confidence interval of 95% and R^2^ values at *p*-value < 0.005 were considered significant.

**Table 1 viruses-11-00578-t001:** (**a**) Day-wise changes observed in the hematological markers post–CHIKV infection in 8 weeks old mice. (**b**) Day-wise changes observed in the hematological markers post–CHIKV infection in 20 weeks old mice.

**(a)**											
**Days of Infection**	**Day 3**		**Day 6**		**Day 9**		**Day 12**		**Day 15**		**Normal Range**
**Parameters**	**Mean**	**Stdev**	**Mean**	**Stdev**	**Mean**	**Stdev**	**Mean**	**Stdev**	**Mean**	**Stdev**	
Hb (g/dL)	15.6	0.62	15.28	0.68	13.88	1.23	15.18	0.28	15.25	0.58	11.2–16.4
TLC (×10^3^/mm^3^)	3.05	0.56	5.25	0.90	6.30	0.54	5.00	1.01	5.00	1.45	1.8–5.2
DLC-neutrophils (%)	18.75	0.96	20.75	2.50	22.25	4.50	19.00	1.41	19.75	3.10	8–20
DLC lymphocytes (%)	75.25	0.96	78.00	2.16	80.25	4.99	75.25	0.96	74.75	3.86	76–91
RBC (×10^6^/mm^3^)	10.78	0.46	10.18	0.36	9.49	0.98	10.08	0.24	10.24	0.65	6.1–10.7
MCV (fL)	47.15	0.67	47.65	1.27	47.35	1.33	47.65	0.62	46.83	2.00	43.4–47.8
MCH (pg)	14.5	0.22	15.00	0.45	14.63	0.35	15.10	0.18	14.90	0.54	14.8–17.6
MCHC (%)	30.75	0.45	26.98	9.35	30.90	1.15	31.63	0.30	31.85	0.35	29.3–35.9
Platelet count (mm^3^)	1097	257.27	1193.00	243.68	770.00	478.09	997.25	299.24	1009.00	407.56	285–890
**(b)**											
**Days of Infection**	**Day 3**		**Day 6**		**Day 9**		**Day 12**		**Day 15**		**Normal Range**
**Parameters**	**Mean**	**Stdev**	**Mean**	**Stdev**	**Mean**	**Stdev**	**Mean**	**Stdev**	**Mean**	**Stdev**	
Hb (g/dl)	14.575	0.359398	14.38	1.06	13.43	0.96	14.05	0.75	15.17	0.81	11.2–16.4
TLC (×10^3^/mm^3^)	4.28025	1.211589	4.76	1.28	5.75	1.45	5.80	1.68	6.95	0.83	1.8–5.2
DLC-neutrophils (%)	12.75	0.732006	45.00	0.17	42.50	0.10	61.25	0.30	53.33	0.49	8–20
DLC lymphocytes (%)	90.275	4.453744	96.38	1.39	91.75	10.50	96.70	1.70	97.47	0.64	76–91
RBC (×10^6^/mm^3^)	9.8675	0.314788	9.68	0.51	8.97	0.55	9.53	0.52	9.66	0.68	6.1–10.7
MCV (fL)	51.075	1.05317	51.60	0.93	50.83	1.32	51.95	1.81	52.70	1.21	43.4–47.8
MCH (pg)	18.8	0.182574	19.83	0.46	20.95	0.31	19.73	0.13	20.20	0.85	14.8–17.6
MCHC (%)	38.925	0.713559	38.73	1.28	39.40	0.50	38.38	1.18	38.90	1.73	29.3–35.9
Platelet count (×10^3^/mm^3^)	897.25	181.9146	907.75	94.89	1021.75	384.15	1102.25	216.78	935.00	137.18	285–890

**Table 2 viruses-11-00578-t002:** Day-wise changes observed in sera-specific hematological markers post–CHIKV infections in 8 weeks and 20 weeks old mice.

DOI	3 dpi (Mean ± Standard Deviation)		6 dpi (Mean ± Standard Deviation)		9 dpi (Mean ± Standard Deviation)		12 dpi (Mean ± Standard Deviation)		15 dpi (Mean ± Standard Deviation)		Average Values in C57BL/6ICGEB Mice	*p*-Value
Age	8 Weeks	20 Weeks	8 Weeks	20 Weeks	8 Weeks	20 Weeks	8 Weeks	20 Weeks	8 Weeks	20 Weeks		
AST/SGOT (U/L)	90.4 ± 5.23	135.95 ± 1.34	116.4 ± 7.58	135.55 ± 4.52	135.8 ± 4.36	257.42 ± 2.36	131.6 ± 5.33	145.51 ± 5.23	134 ± 6.3	140.65 ± 5.89	62.21–87.7	>0.05
ALT/SGPT (U/L)	37.2 ± 4.54	94.00 ± 8.48	47 ± 9.3	71.1 ± 11.15	38.6 ± 10.28	76.46 ± 5.68	64.2 ± 9.18	78.85 ± 8.56	50.2 ± 10.19	80.24 ± 9.65	23.18–30.82	>0.05
ALP (U/L)	297 ± 5.25	230.2 ± 2.58	295 ± 8.28	280 ± 6.28	294 ± 11.19	300 ± 2.19	248 ± 5.34	305 ± 6.34	304 ± 5.23	318 ± 4.23	35–96	>0.05
RF	Absent	Absent	Absent	Absent	Absent	Absent	Absent	Present (50%)	Absent	Present (50%)	Not applicable	

**Table 3 viruses-11-00578-t003:** Levels of various cell–mediated immunity markers at stages of disease progression.

Cytokines and Chemokines (Semi Quantitative Expression with Respect to Controls)				
Age Group	8 Weeks	8 Weeks	20 Weeks	20 Weeks
Days post infections (dpi)	Day 1–Day 6	Day 7–Day 15	Day 1–Day 6	Day 7–Day 15
Clinical Manifestations	Asymptomatic Acute	Post-Acute	Severe Acute	Post-Acute
IFN-α	+	++	++	+++
CCR1			++	-
CCR2			++	+
BST-2	++		+	+
TNF-α	++	+	+++	+
VEGF	+		++	
IL-4		++		++
GM-CSF		+		+
CCL2(MCP-1)		+		++
RANTES				+
RANKL		+		++
IL-1β	+	+	++	+++
IL-6	+	++	+++	+++

Note: + 1× higher than uninfected control; ++ 2× higher than uninfected control; +++ 3× higher than uninfected control.

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
