# Peer review of "Establishment and Comparison of Pathogenicity and Related Neurotropism in Two Age Groups of Immune Competent Mice, C57BL/6J Using an Indian Isolate of Chikungunya Virus (CHIKV)"

_viruses, 2019, doi:10.3390/v11060578_

Reviewer 1 Report

This article point at an important issue of the Chikungunya virus disease that is the influence of age on disease severity, and the use of immunocompetent adult mice is of relevance to the topic.

However, the article needs to be improved since the way the results are presented is not convincing and some conclusions should be revised.

A general comment applying to all the paper, as it is presented, is that figures can be better organized, statistical analyses and number of replicates or infected animals must be clearly presented, and some data should not be repeated in more than one figure. As a major point, histopathological analyses are not convincing and I suggest to merge sections on viral replication in brain and joints.

As for the limitations of the study, the authors finely discussed it in the conclusions.

Point 1) Some definitions, as well as article introduction need rephrasing.

In particular the authors ill define the acute and post-acute phase of CHIKD as they can also read in the paper they cite (ref 4 line 35). Indeed, in humans the acute phase last 21 days and it is not defined as the pyretic phase. This definition should, thus, be corrected. Moreover, the illness progress in mice can be different.

So, the authors can either use early versus late acute phase or better justify how they chose to set at day 7 the limit between acute and post acute phase.

 Line 45: I think disease severity correlates patient’s age in all the phases of the disease (including the acute phase).

Lines 46-47 and 49-50 (and 213-215): You stated that relevant models of CHIKD are lacking but I do not feel the same. I think you should comment on Non Human Primate models (see works by P. Roques and collaborators or Ziegler et al 2010 or review by Broeckel PNTD 2017).

Lines 51 to 66: please rephrase the paragraphs and correct punctuation (a, b,c, shouldn’t be separated in different and distant sentences)

Point 2) Methods

Line 100: please spell CMC

Line 106 and all over the text: I suggest to write viral RNA or vRNA instead of VNA

Line117: a more precise definition of the parameters used to assess the morbidity score could be useful especially considering your conclusions on disease correlation to brain and joint invasion. Are neurological signs considered? Did you noticed differences in specific clinical signs among older and younger mice?

Line 110-120: please indicate the number of animals tested in each group cause it is relevant to understand the significativity of your findings.

Line 121-131 (and result sections) Did each group of animals consist of 4 mice (2 males, 2 female). I meand are data coming from 4 animals/time point in each age group?

Please clarify it indicating constantly the number of animals in figures.

Line 175-186 and results section on cell mediated immunity: Why did you choose to measure some cytokines in splenocytes and other in sera?

Line 180: replace was with were (data were)

Line 195: please write “presence of viral antigens in joint and brain tissues” instead of viral particles

Line 195-196: how can you conclude that the damage is caused by viral particles?

Line 197-211 on statistical analyses (also apply to results and figures): I have some doubts on the use of a parametric student t test in a study taking into consideration a low number of animals. Independently on the normality test used (which one?), the number of tested individuals probably requires a non parametric tes.

Moreover, for correlation tests I think spearman test is more appropriate.

Please remember to name the test used in figure legends, indicate the number of individuals and the P values obtained in graphs.

Point3) Results and discussion

Line 223: revise punctuation.

Line 234-237: I do not see the results comparing 8 to 20 weeks old mice. Suppl fig 1 only shows 8 weeks old animals. Moreover, you stated the chik strain 1 is the more pathogenic when it is actually the most suitable (strain 6 kills all the animals, thus is more virulent and pathogenic).

Please indicate the number of animals tested.

Suppl fig 1: Please indicate the number of animals tested and day post infection.

Line 256 and figure1: is the intermittent drop significant?

Please graph the single individuals ad separated dots.

The morbidity score limit in graph 1A and 1B doesn’t correspond to limit define in results text (5).

Line 263 and figure 1: please indicate the statistical test used and if you show mean or median values

Figure 1 legend: please be careful to indicate every thing you show (mean/media, limit of detections, statistical test?) and correct x-axes in 1D, indicate statistics in graphs if significant.

Line 280: spell DLC

Lines 288-289: what do you mean as most animals showed enlarged spleen, how many? 3 out of 4?

Table 1: any statistical test was ran? Are the values indicated mean values?

Line 305: cancel “post” it is already in d.p.i.

Table 2: is it mean +/- SD shown ? Please indicate it.

Line 318: better say replicating virus than viral particles

Fig 2: please indicate if mean or median SD etc are indicated, indicate the number of animal tested, the statistics

Line 333: we miss a table, that can go in supplementary, showing survival of animals in each group, we learn here that some mice died during the infection time laps (how many?)

Line 337-338: “was observed” is repeated twice.

Fig 3: please split the graph in at least 2 separeted graphs (younger and older) or divide it per tissue, as it is the graph is confusing. Indicate statistical significativity, if present, and viral RNA copies are measure per gram of tissue or other factors? Indicate it in y-axis.

Section “morbidity in mice correlates with viral replication in brain”:

This part is less convincing and some concepts and results presented before are here reported again:

A)      Is the replication in brain the only correlating to morbidity?

B)      Are neurological parameters taken into specific consideration?

C)      Are Data in figure 4A the same presented in Figure 1 ?

D)      Limit (horizontal line) of morbidity score is different from figure 1 (adjust one of them properly)

E)      The immunohistology is not convincing, If the authors want to include this result, they have to choose another picture, ameliorate resolution, show not infected and 8 weeks old mice results, or repeat the staining.

F)       Line 374-376, you name viral replication in astrocytes, please consider that to define cell tropism you should carry out deep morphological analyses and provide a double staining of viral antigens and specific cell markers

G)     In figure legend some parameters are not defined (arrows etc)

H)     Line 378: please provide evidence of kinetics of viral replication and morbidity in suppl fig or name the figure in which we can see it.

Section on Arthralgia persistence: this part is also less convincing but can be easily improved:

Fig 5a: looks like a repetition of data presented in figure 1

Fig5d: please provide better pictures and not infected and 8 weeks old derived analyses together with results in olderinfected mice. Remember to define arrows and circles in fig legend.

Please provide statistical analyses for the data presented in fig 5.

Line 415: have instead of has

Line 420 were instead of was

Line 430: IFN indicates alpha, beta or gamma? Which tissues?

Table4: data should be presented differently, haven’t you concentration or OD or MFI values to display?

Fig 6 and 7: I would replace fig 6B and C; and 7 B and C with a comparison of neutralizing ab in younger versus older mice.

Point 4) Conclusions

I suggest to rephrase and revise punctuation when presenting a), b) and c) points (the same as in the introduction.

Line 497 point c) is less convincing and line 508-510 the conclusion on macrophages/monocytes should be milder.

Lines 513-514 point d: to be revised cause histopathological data are not indicative if not replaced.

Author Response

Reviewer 1

This article points at an important issue of the Chikungunya virus disease that is the influence of age on disease severity, and the use of immunocompetent adult mice is of relevance to the topic.

However, the article needs to be improved since the way the results are presented is not convincing and some conclusions should be revised.

A general comment applying to all the paper, as it is presented, is that figures can be better organized, statistical analyses and number of replicates or infected animals must be clearly presented, and some data should not be repeated in more than one figure. As a major point, histopathological analyses are not convincing, and I suggest merging sections on viral replication in brain and joints.

As for the limitations of the study, the authors finely discussed it in the conclusions.

Response: We thank the reviewers for their intense review of our manuscript. This has helped in shaping our manuscript much better. The figures have been better organized, the statistical analysis and the number is replicates/infected animals in each group has been represented. Further, interpretation of data has been better represented in the revision and in the figure legends now. The section of viral replication in brain and joints has also now been merged.

Point 1) Some definitions, as well as article introduction need rephrasing.

In particular the authors ill define the acute and post-acute phase of CHIKD as they can also read in the paper they cite (ref 4 line 35). Indeed, in humans the acute phase last 21 days and it is not defined as the pyretic phase. This definition should, thus, be corrected. Moreover, the illness progress in mice can be different.

So, the authors can either use early versus late acute phase or better justify how they chose to set at day 7 the limit between acute and post acute phase.

Response: The following statement has been corrected, line 37-40

 Line 45: I think disease severity correlates patient’s age in all the phases of the disease (including the acute phase).

Response: This has been corrected

Lines 46-47 and 49-50 (and 213-215): You stated that relevant models of CHIKD are lacking but I do not feel the same. I think you should comment on Non Human Primate models (see works by P. Roques and collaborators or Ziegler et al 2010 or review by Broeckel PNTD 2017).

Response: We have rephrased to paragraph for better understanding; line 46-54

Lines 51 to 66: please rephrase the paragraphs and correct punctuation (a, b,c, shouldn’t be separated in different and distant sentences)

Response: We have rephrased the paragraph so as to make it one complete paragraph in the revised manuscript. 

Point 2) Methods

Line 100: please spell CMC

Response: We have made the necessary changes in the revised MS.

Line 106 and all over the text: I suggest to write viral RNA or vRNA instead of VNA

Response: We have modified the manuscript with the relevant terminology.

Line117: a more precise definition of the parameters used to assess the morbidity score could be useful especially considering your conclusions on disease correlation to brain and joint invasion. Are neurological signs considered? Did you noticed differences in specific clinical signs among older and younger mice?

Response: Ref 41 talks about the specific signs considered for the deduction of morbidity. Mice were observed and given a numerical score for behavioral signs of encephalitis (0, no detectable sign of disease; 1, ruffled fur; 2, slightly hunched back and ruffled fur; 3, very hunched back and lethargy; and 4, death) as well along with other features dealing with the overall health of mice. The signs of viral encephalitis were age dependent as well as dose dependent. The main characteristics taken under consideration have been included in the main text in the revised manuscript (line 137-140).

Line 110-120: please indicate the number of animals tested in each group cause it is relevant to understand the significativity of your findings.

Response: The number of animals used for the various studies have now been included and more clearly mentioned in the manuscript.

Line 121-131 (and result sections) Did each group of animals consist of 4 mice (2 males, 2 female). I meand are data coming from 4 animals/time point in each age group?

Please clarify it indicating constantly the number of animals in figures.

Response: Yes, the data is coming from 4 mice each group. It is now mentioned in the figure legend wherever applicable.

Line 175-186 and results section on cell mediated immunity: Why did you choose to measure some cytokines in splenocytes and other in sera?

Response: The choice of test for cytokines and chemokines either in splenocytes or sera was based on the literature review. The details have been discussed in the manuscript where relevant. Also, it is better to check the levels of certain chemokines/cytokines in the sera samples especially during the post-acute phase of the disease as splenocytes is the not the primary site of infection in chikungunya.

Line 180: replace was with were (data were)

Response: We have made the necessary changes in the revised MS.

Line 195: please write “presence of viral antigens in joint and brain tissues” instead of viral particles

Response: We have made the necessary changes in the revised MS.

Line 195-196: how can you conclude that the damage is caused by viral particles?

Response: We apologize for this overstatement. We have made the necessary changes in the revised MS.

Line 197-211 on statistical analyses (also apply to results and figures): I have some doubts on the use of a parametric student t test in a study taking into consideration a low number of animals. Independently on the normality test used (which one?), the number of tested individuals probably requires a non-parametric test.

Moreover, for correlation tests I think spearman test is more appropriate.

Please remember to name the test used in figure legends, indicate the number of individuals and the P values obtained in graphs.

Response:  We thank the reviewers for their suggestion, the statistical analysis have been improved. We also performed the Spearman test for the correlation analysis and could not find any differences between the two.

Point3) Results and discussion

Line 223: revise punctuation.

Response: We have made the necessary changes in the revised MS.

Line 234-237: I do not see the results comparing 8 to 20 weeks old mice. Suppl fig 1 only shows 8 weeks old animals. Moreover, you stated the chik strain 1 is the more pathogenic when it is actually the most suitable (strain 6 kills all the animals, thus is more virulent and pathogenic).

Please indicate the number of animals tested.

Response:  The supplementary figure 1 describes results of both 8 weeks and 20 weeks old mice; it may be due to the format change that the figure might not be properly visible. We request the reviewer to check the word file of the manuscript, as there are some format errors in the PDF version. We thank the reviewers for pointing this out. The aim of the study was not to look for animals’ mortality (as this is not the humane end-point of the disease) which is why we did not choose strains that were most pathogenic. We have made the necessary corrections regarding the choice of strain throughout the revised manuscript. We aimed to study the age-related differences in the host response in the acute and post-acute phase of the disease.

Suppl fig 1: Please indicate the number of animals tested and day post infection.

Response: The legend for Suppl fig 1 is now improved

Line 256 and figure1: is the intermittent drop significant? Please graph the single individual’s ad separated dots.

Response: When compared to the results observed at 15 dpi, the intermittent drop at 12 dpi was significant when the animals were infected with 1x106 and 1x108 virus particles. But, when compared to 9 dpi the drop was not significant. Thus, the result was left unconcluded. We have now represented the graph of the individual animals as data points.

The morbidity score limit in graph 1A and 1B doesn’t correspond to limit define in results text (5).

Response: We thank the reviewers for pointing this out, the limits lines were included manually, and this error occurred while resizing the image in Graphpad. We sincerely apologise for this mistake and this had now been corrected. 

Line 263 and figure 1: please indicate the statistical test used and if you show mean or median values

Response: The statistical test used have been indicated, we are depicting the mean values with standard deviation

Figure 1 legend: please be careful to indicate everything you show (mean/media, limit of detections, statistical test?) and correct x-axes in 1D, indicate statistics in graphs if significant.

Response: The figures have been corrected

Line 280: spell DLC

Response: We have made the necessary changes in the revised MS.

Lines 288-289: what do you mean as most animals showed enlarged spleen, how many? 3 out of 4?

Response: We apologize for this. Exact values is now added to the manuscript, line 302.

Table 1: any statistical test was ran? Are the values indicated mean values?

Response: No statistical test was done for this and the values indicated in the table are the mean values with standard deviation.

Line 305: cancel “post” it is already in d.p.i.

Response: We have made the necessary changes in the revised MS.

Table 2: is it mean +/- SD shown ? Please indicate it.

Response: We have made the necessary changes in the revised MS.

Line 318: better say replicating virus than viral particles

Response: We have made the necessary changes in the revised MS.

Fig 2: please indicate if mean or median SD etc are indicated, indicate the number of animal tested, the statistics

Response: The legend has been corrected, but it is very difficult to indicate the statistics done in the figure, hence we have not included in the figure.

Line 333: we miss a table, that can go in supplementary, showing survival of animals in each group, we learn here that some mice died during the infection time laps (how many?)

Response: 2 animals (both female) died at 15dpi.  We did not make a separate table as percentage of alive animals are depicted in the table itself.

Line 337-338: “was observed” is repeated twice.

Response: We have made the necessary changes in the revised MS.

Fig 3: please split the graph in at least 2 separeted graphs (younger and older) or divide it per tissue, as it is the graph is confusing. Indicate statistical significativity, if present, and viral RNA copies are measure per gram of tissue or other factors? Indicate it in y-axis.

Response: We have now color coded the figure for better clarity. We tried making two graphs for 8 weeks and 20 weeks but the comparison did not come out well and so we resorted to a single graph. We agree the graph is data heavy but the gist of the results is best represented in this way. We have elaborated the findings more clearly in the main text.

Section “morbidity in mice correlates with viral replication in brain”:

This part is less convincing and some concepts and results presented before are here reported again:

A)     Is the replication in brain the only correlating to morbidity?

Response: Yes, only replication of virus in brain was significantly correlating to the morbidity

B)      Are neurological parameters taken into specific consideration?

Response: Viral encephalitis parameters were taken into considerations while calculating the morbidity of animals

C)      Are Data in figure 4A the same presented in Figure 1 ?

Response: Figure 1 talks about the morbidity as various virus doses, whereas Figure 4A depicts day-wise age group difference when infected with 1x106 viruses.

D)      Limit (horizontal line) of morbidity score is different from figure 1 (adjust one of them properly)

Response: There was a mistake in figure 1 and that has now been corrected, we thank the reviewer for pointing this out

E)      The immunohistology is not convincing, If the authors want to include this result, they have to choose another picture, ameliorate resolution, show not infected and 8 weeks old mice results, or repeat the staining.

Response: The same has been removed from the paper, we are working on this aspect in detail and decided that it would be better if it goes with the other manuscript with more details for both the aspects.

F)       Line 374-376, you name viral replication in astrocytes, please consider that to define cell tropism you should carry out deep morphological analyses and provide a double staining of viral antigens and specific cell markers

Response: We understand the claim made in this section, as no specific markers were used for the study and the claims were based on physical observations; thus the same has been corrected in the revised manuscript.

G)     In figure legend some parameters are not defined (arrows etc)

Response: The figure has been removed and is not a part of the manuscript anymore

H)     Line 378: please provide evidence of kinetics of viral replication and morbidity in suppl fig or name the figure in which we can see it.

Response: Figs 1, 2 and 3 represent kinetics of viral replication either as plaque forming units or viral RNA. Fig 5 discusses the correlation between morbidity and viral replication in brain specifically.

Section on Arthralgia persistence: this part is also less convincing but can be easily improved:

Fig 5a: looks like a repetition of data presented in figure 1

Response: Figure 1 talks about the changes in limb thickness as various virus doses, whereas Figure 5A depicts day-wise age group difference when infected with 1x106 viruses.

Fig5d: please provide better pictures and not infected and 8 weeks old derived analyses together with results in olderinfected mice. Remember to define arrows and circles in fig legend.

Response: The figure is not a part of the manuscript anymore

Please provide statistical analyses for the data presented in fig 5.

Response: The data has been incorporated in the revised MS.

 Line 415: have instead of has

Response: We have made the necessary changes in the revised MS.

Line 420 were instead of was

Response: We have made the necessary changes in the revised MS.

Line 430: IFN indicates alpha, beta or gamma? Which tissues?

Response:  We have made the necessary changes in the revised MS, line 448

Table4: data should be presented differently, haven’t you concentration or OD or MFI values to display?

Response:  We have used two different platforms for cytokine and chemokines profiling, that is why to we have represented it in the qualitative manner, but the data from two different technologies was normalised with respect to the uninfected mice and the data was interpreted as fold change differences with respect to the uninfected mean. We can provide the analysed raw data of the FACS for your consideration but it is a 52 pages document and did not know how to add to the manuscript. We will be happy to share with you if you wish.

Fig 6 and 7: I would replace fig 6B and C; and 7 B and C with a comparison of neutralizing ab in younger versus older mice.

Response: We thank the reviewers for this suggestion; we have included this information in suppl figure 3.

Point 4) Conclusions

I suggest to rephrase and revise punctuation when presenting a), b) and c) points (the same as in the introduction.

Response: While we have changed the introduction, we feel having a point wise conclusion brings out the significant findings of the work better. We would prefer to leave it as such if the reviewer and the editorial committee allow.

Line 497 point c) is less convincing and line 508-510 the conclusion on macrophages/monocytes should be milder.

Response: We have made the necessary changes in the revised MS.

Lines 513-514 point d: to be revised cause histopathological data are not indicative if not replaced.

Response: We have made the necessary changes in the revised MS.

Reviewer 2 Report

The present manuscript provides a broad description of CHIKV interactions and impacts in the vertebrate host model aiming to clarify the differences regarding the consequences when hosts have distinct ages. The article is interesting and brings important information about CHIKV pathogenesis. Although the manuscript is long and could be shortened with the main findings, it is important and I consider it accepted after a few modifications, described below:

Minor reviews:

1-Line 223 – “…associated with CHIK We show…”, change to “…associated with CHIK. We show…”;

2-Line 227 – “distinct difference” is not redundant? 

3-Figure 3 - Graphically, figure 3 is not clear. The bars are almost overlapped and the colors are not favoring the interpretation. A new representation must be presented. 

4-Figure 4D - The figure resolution is low, it is really hard to see anything. Also, the legend does not describe the arrows and the marker we should observe (red is....).

5-Figure 5D - The figure resolution is also low. There is no description of the red circles and arrows in the legend. 

6-Figure 4D and 5D would include the relative non-infected controls at least and 8 weeks counterpart. Why the authors did not include them by effect of comparison?

7- The authors injected the mice with the defined concentrations described in the methods. How  the determined  optimum dose would be compared with the titers a human host usually needs to be inoculated by a mosquito to be infected? And the titers usually described in symptomatic patients? A discussion about this correlations would be interesting for the readers.

Author Response

Reviewer 2

The present manuscript provides a broad description of CHIKV interactions and impacts in the vertebrate host model aiming to clarify the differences regarding the consequences when hosts have distinct ages. The article is interesting and brings important information about CHIKV pathogenesis. Although the manuscript is long and could be shortened with the main findings, it is important and I consider it accepted after a few modifications, described below:

Minor reviews:

1-Line 223 – “…associated with CHIK We show…”, change to “…associated with CHIK. We show…”;

Response: We have made the necessary changes in the revised MS.

2-Line 227 – “distinct difference” is not redundant? 

Response: We have made the necessary changes in the revised MS.

3-Figure 3 - Graphically, figure 3 is not clear. The bars are almost overlapped and the colors are not favoring the interpretation. A new representation must be presented. 

Response: We have now color coded the figure

4-Figure 4D - The figure resolution is low, it is really hard to see anything. Also, the legend does not describe the arrows and the marker we should observe (red is....).

Response: The result section has been revised and the figures have been removed as suggested by Reviewer 1

5-Figure 5D - The figure resolution is also low. There is no description of the red circles and arrows in the legend. 

Response: The result section has been revised and the figures have been removed as suggested by Reviewer 1

6-Figure 4D and 5D would include the relative non-infected controls at least and 8 weeks counterpart. Why the authors did not include them by effect of comparison?

Response: The result section has been revised and the figures have been removed as suggested by Reviewer 1

7- The authors injected the mice with the defined concentrations described in the methods. How the determined  optimum dose would be compared with the titers a human host usually needs to be inoculated by a mosquito to be infected? And the titers usually described in symptomatic patients? A discussion about this correlations would be interesting for the readers.

Response: The amount of virus that is injected into a human by a mosquito bite is very different from an injection based infection model and it will be unwise to compare the two. Had we done the infection using a mosquito system where we allow infected mosquitoes to bite the mice and thereby infect the mice, then it would be make a good discussion point. However, this was not the approved method in our protocol and so we couldn't use this method for infecting the mice. Also, using this method, it would be difficult to accomplish the varied concentrations that was part of the study.

Reviewer 3 Report

Summary: The authors evaluate pathogenesis of several strains belonging to the ECSA genotype derived from clinical isolates in adult C57BL/6J mice model, and then analyzed the strain that was most pathogenic for dose-dependent pathogenesis, focusing on acute and post-acute phase of infection in two age groups of mice that are either eight or 20 weeks old mice groups. They measure several disease progression attributes correlated to morbidity and linked neurotropism and limb thickness in the two age groups.

Critics:

1) Although it is recognized that aging has an impact on immune system response (PMID: 29242543), based on a literature review of age groups in humans, I’m afraid that the age groups that the authors chose (young=15-18 and old=40-45) to reflect difference of human immune system, might be considered equal or overlapping (PMID: 26702035). Groups that revealed an age-associated immune system difference that young are considered of age 21–30 and older with age ≥65 subjects, as well as when young are considered<40 while="" older="" are="">70 (PMID: 20667703). Another study (PMID:19596035) compared young (21–40) and elderly (64–92). Panda et al carried out a study of TLR-induced cytokine production in young (21–30) and older (≥65) individuals (PMID: 20100933). It seems like the cut-off is less than 40 and more than 64 to find group age related immune system differences.

Moreover, in my opinion, the age difference between the mice is not large enough to cause a change in immunological response. Mice that are 6 weeks old are considered as adults, therefore difference in the immune response would be found with mice that are younger or older than 6 week. Here the groups are 8 or 20 weeks old, so both adults groups.

I would ask the authors to find references in the literature that show that these age groups are relevant both for human and mouse as far as age difference related to immune system changes.

Can authors add an older group of mice to detect difference in immune system response when mice age is similar to human >65?

2) Have the 15 strains that authors used been shown to be neurotropic in human? What is the genomic difference among the strains? Which genes present mutations?

3) Please make more clear in both panels of Figure 1 what is the x axis. It is not really clear if it is PFUs or time. If it is PFUs it should not be presented as continuous line but as histogram. What is the DL50?

4) Why authors only used CHIKV#01? Authors should have also performed experiments with #11, 12 and 20 that are similar to #01, with #6 since it is highly virulent focusing on differences in early infection (day1-6), and also add an “attenuated” group using some strains such #32, 40 and 42

5) Figure 4: combine panel B and C so that the reader can assess if there is any difference between the age-groups or make y-axis scale equal so that is comparable. Please add an additional panel that shows immunohistopathology of the brain tissues of CHIKV infected 8 weeks old mice.

6) Figure 5: Please add an additional panel that shows histopathology of the tissues of CHIKV infected 8 weeks old mice.

7) Please give quantitative expression for cytokines and chemokines.

Author Response

Reviewer 3

Summary: The authors evaluate pathogenesis of several strains belonging to the ECSA genotype derived from clinical isolates in adult C57BL/6J mice model, and then analyzed the strain that was most pathogenic for dose-dependent pathogenesis, focusing on acute and post-acute phase of infection in two age groups of mice that are either eight or 20 weeks old mice groups. They measure several disease progression attributes correlated to morbidity and linked neurotropism and limb thickness in the two age groups.

Critics:

1)Although it is recognized that aging has an impact on immune system response (PMID: 29242543), based on a literature review of age groups in humans, I’m afraid that the age groups that the authors chose (young=15-18 and old=40-45) to reflect difference of human immune system, might be considered equal or overlapping (PMID: 26702035). Groups that revealed an age-associated immune system difference that young are considered of age 21–30 and older with age ≥65 subjects, as well as when young are considered<40 while="" older="" are="">70 (PMID: 20667703). Another study (PMID:19596035) compared young (21–40) and elderly (64–92). Panda et al carried out a study of TLR-induced cytokine production in young (21–30) and older (≥65) individuals (PMID: 20100933). It seems like the cut-off is less than 40 and more than 64 to find group age related immune system differences.

Moreover, in my opinion, the age difference between the mice is not large enough to cause a change in immunological response. Mice that are 6 weeks old are considered as adults, therefore difference in the immune response would be found with mice that are younger or older than 6 week. Here the groups are 8 or 20 weeks old, so both adults groups.

I would ask the authors to find references in the literature that show that these age groups are relevant both for human and mouse as far as age difference related to immune system changes.

Response: PMID: 26702035 talks about the strength of B cell and the T cell response in this age group, whereas if the reviewer notices the number of affected individuals due to the Pandemic influenza is in this the age group of 15-50 years old, thus for the analysis of the viral disease and its pathogenesis of the two extremes of the most affected age group was considered.

PMID: 20667703, We understand that in this study the age groups considered for the study are as the same as mentioned in your comment. The review mentioned data of the patients that were<40 and="">70. But in the study cited by the authors this statement (PMID: 17202359), only two groups have been considered and if you closely analyse the data there is huge variations within the young population.

In this study, our aim is to understand the age-related response in the most affected population starting age 15yrs with a median at 40 when infected with the 2010 Delhi CHIKV strain (PMID: 28379375), rather than commenting on the immune response of young vs old. Thus, we have taken the age group

PMC5367550 talks about the discrepancies in the mice data, as most labs only use murine model in age range of 6-12 weeks with the upper limit in 20 weeks old animals, irrespective of the disease studied. Therefore, for this study we decided to use the animals that most closely resembles the human infection age range during CHIKV infections as most information in the Indian scenario is available from this age group.

Can authors add an older group of mice to detect difference in immune system response when mice age is similar to human >65?

Response: This was not the scope of the study approved by IAEC, thus this data cannot be generated.

2) Have the 15 strains that authors used been shown to be neurotropic in human? What is the genomic difference among the strains? Which genes present mutations?

Response: We are in the process of sequencing the whole genome of the clinical strains and the results are part of a separate manuscript.

3) Please make more clear in both panels of Figure 1 what is the x axis. It is not really clear if it is PFUs or time. If it is PFUs it should not be presented as continuous line but as histogram. What is the DL50?

Response: MLD50 is already defined in supplementary figure 1. In figure 1 the PFU dependent morbidity and change in the limb thickness.

In Figure 1a and 1b x-axis is the days post infection and in figure 1c and 1d x-axis is PFU which is already represented as histograms.

4) Why authors only used CHIKV#01? Authors should have also performed experiments with #11, 12 and 20 that are similar to #01, with #6 since it is highly virulent focusing on differences in early infection (day1-6), and also add an “attenuated” group using some strains such #32, 40 and 42

Response: The focus of the current study was to study impact of CHIK infection on different age groups and we chose the most optimal and representative virus to study this aspect. Studying the pathogenicity patterns of the several strains will be more informative once the genetic information is available and will be the scope a complete study by itself.

5) Figure 4: combine panel B and C so that the reader can assess if there is any difference between the age-groups or make y-axis scale equal so that is comparable. Please add an additional panel that shows immunohistopathology of the brain tissues of CHIKV infected 8 weeks old mice.

Response: This has been done in the revised MS. We have removed the histopathology data as advised by another reviewer and the whole sectioned have been merged along with the replication kinetics

6) Figure 5: Please add an additional panel that shows histopathology of the tissues of CHIKV infected 8 weeks old mice.

 Response: We have removed the histopathology data as advised by another reviewer and the whole sectioned have been merged along with the replication kinetics

7) Please give quantitative expression for cytokines and chemokines.

Response: We have used two different platforms for cytokine and chemokines profiling, that is why to we have represented it in the qualitative manner, but the data from two different technologies was normalised with respect to the uninfected mice and the data was interpreted as fold change differences wrt the uninfected mean. We can provide the analysed raw data of the FACS for your consideration but it is a 52 pages document and did not know how to add to the manuscript. We will be happy to share with you if you wish.

Round  2

Reviewer 1 Report

The revised manuscript is greatly improved in terms of clarity.

I just recommend a careful proofreading beacause few typos are still present in the text.

Author Response

The revised manuscript is greatly improved in terms of clarity.

I just recommend a careful proofreading beacause few typos are still present in the text.

Response: We have done a thorough proof reading and also given the manuscript to our in-house copyediting department for their inputs. We hope that the manuscript is now in proper order.

Reviewer 3 Report

Please add to the introduction the discussion about age groups that has been produced through comment 1) as I did ask through my comment and track changes so that I can see where this has been added, and please state clearly in the text the goal of the project.

[1)Although it is recognized that aging has an impact on immune system response (PMID: 29242543), based on a literature review of age groups in humans, I’m afraid that the age groups that the authors chose (young=15-18 and old=40-45) to reflect difference of human immune system, might be considered equal or overlapping (PMID: 26702035). Groups that revealed an age-associated immune system difference that young are considered of age 21–30 and older with age ≥65 subjects, as well as when young are considered<40 while="" older="" are="">70 (PMID: 20667703). Another study (PMID:19596035) compared young (21–40) and elderly (64–92). Panda et al carried out a study of TLR-induced cytokine production in young (21–30) and older (≥65) individuals (PMID: 20100933). It seems like the cut-off is less than 40 and more than 64 to find group age related immune system differences. Moreover, in my opinion, the age difference between the mice is not large enough to cause a change in immunological response. Mice that are 6 weeks old are considered as adults, therefore difference in the immune response would be found with mice that are younger or older than 6 week. Here the groups are 8 or 20 weeks old, so both adults groups. I would ask the authors to find references in the literature that show that these age groups are relevant both for human and mouse as far as age difference related to immune system changes. Response: PMID: 26702035 talks about the strength of B cell and the T cell response in this age group, whereas if the reviewer notices the number of affected individuals due to the Pandemic influenza is in this the age group of 15-50 years old, thus for the analysis of the viral disease and its pathogenesis of the two extremes of the most affected age group was considered. PMID: 20667703, We understand that in this study the age groups considered for the study are as the same as mentioned in your comment. The review mentioned data of the patients that were<40 and="">70. But in the study cited by the authors this statement (PMID: 17202359), only two groups have been considered and if you closely analyse the data there is huge variations within the young population. In this study, our aim is to understand the age-related response in the most affected population starting age 15yrs with a median at 40 when infected with the 2010 Delhi CHIKV strain (PMID: 28379375), rather than commenting on the immune response of young vs old. Thus, we have taken the age group. PMC5367550 talks about the discrepancies in the mice data, as most labs only use murine model in age range of 6-12 weeks with the upper limit in 20 weeks old animals, irrespective of the disease studied. Therefore, for this study we decided to use the animals that most closely resembles the human infection age range during CHIKV infections as most information in the Indian scenario is available from this age group.]

Please change the coloring of figure 3, it is very not distinguishable if you are color-blind and avoid use of pattern as does not make any more clear the difference in groups, especially if the bars are so thin. When printed, this figure is not understandable. The figure can be improved by showing it horizontal rather than vertical and make every bar thicker.

Author Response

Please add to the introduction the discussion about age groups that has been produced through comment 1) as I did ask through my comment and track changes so that I can see where this has been added, and please state clearly in the text the goal of the project.

Response: While we appreciate the reviewer’s comments with respect to the age correlation to age dependent immune status of the mice group we have taken in the study, we would like to emphasize the logic of having taken the two age groups. Our study aims to encapsulate disease condition in the median age group as seen in the human populations during CHIKV outbreaks, ie, 40-45 human years. The study was not designed to study CHIKV pathogenesis in young vs geriatric age groups, as the reviewer has assumed. Several studies have emphasized that 8 weeks mice corresponds to young adult mice with the biological adulthood age corresponding to 15-18 years. The 20 week old mice correspond to human adulthood age of 45-55 years and can be considered as mature adults, which was the reason for us to choose this age group. Furthermore, previous studies have used this age group in studies involving types of arthritis; another reason for us to decide on this age group to study CHIK in totality. These aspects are now been included in the introduction of the MS with relevant refs and are in track mode for easy accessibility. 

Please change the coloring of figure 3, it is very not distinguishable if you are color-blind and avoid use of pattern as does not make any more clear the difference in groups, especially if the bars are so thin. When printed, this figure is not understandable. The figure can be improved by showing it horizontal rather than vertical and make every bar thicker.

Response:

We have redone the figure to bring out the information more clearly. We hope that this edit is up to the expectation of the reviewer and the journal. We thank the reviewer for his efforts to make our MS a good quality research paper.